# Exact exchange-correlation potentials from ground-state electron densities

Bikash Kanungo[1], Paul M. Zimmerman [iD] [2] & Vikram Gavini [iD] [1,3]*

The quest for accurate exchange-correlation functionals has long remained a grand challenge in density functional theory (DFT), as it describes the many-electron quantum mechanical behavior through a computationally tractable quantity—the electron density—without resorting to multi-electron wave functions. The inverse DFT problem of mapping the ground-state density to its exchange-correlation potential is instrumental in aiding functional development in DFT. However, the lack of an accurate and systematically convergent approach has left the problem unresolved, heretofore. This work presents a numerically robust and accurate scheme to evaluate the exact exchange-correlation potentials from correlated ab-initio densities. We cast the inverse DFT problem as a constrained optimization problem and employ a finite-element basis—a systematically convergent and complete basis—to discretize the problem. We demonstrate the accuracy and efficacy of our approach for both weakly and strongly correlated molecular systems, including up to 58 electrons, showing relevance to realistic polyatomic molecules.

[1] Department of Mechanical Engineering, University of Michigan, Ann Arbor, Michigan 48109, USA. [2] Department of Chemistry, University of Michigan, Ann Arbor, Michigan 48109, USA. [3] Department of Materials Science and Engineering, University of Michigan, Ann Arbor, Michigan 48109, USA. *email: vikramg@umich.edu

**D**ensity functional theory (DFT)[1,2] is an essential method for describing electronic states in all manner of nanoscale phenomena, including chemical bonds in molecules, band structures of materials, electron transfer, and reactive metal clusters of proteins. In principle, an exact theory, DFT in practice[3–8], has remained far from exact due to the unavailability of exact exchange-correlation (xc) potentials ($v_{xc}$), which are responsible for describing the quantum mechanical behavior of electrons. Fortunately, $v_{xc}$ is a unique functional of the electron density ($\rho(\mathbf{r})$), so there exists a one-to-one relationship from $v_{xc}(\mathbf{r})$ to $\rho(\mathbf{r})$ and vice versa. This observation presents a possible route forward to construct accurate xc functionals via the transformation of the electron density into $v_{xc}(\mathbf{r})$ through the so-called inverse DFT problem[9–13] (refer to the schematic in Fig. 1). The inverse problem not only provides a route for finding the sole unknown quantity in DFT, it is also central for describing quantum mechanics without resorting to complicated multi-electron wave functions.

Given the large importance of this problem, there have been several attempts to solve the inverse DFT problem, employing either iterative updates[10,11,14–16] or constrained optimization approaches[9,12,17,18]. However, these approaches have suffered from ill-conditioning, thereby resulting in non-unique solutions or causing spurious oscillations in the resultant $v_{xc}(\mathbf{r})$. This ill-conditioning has been largely attributed to the incompleteness of the Gaussian basis sets that were employed to solve the inverse DFT problem[18–20]. Recent efforts[21–23] have presented a different approach, which utilizes the two-electron reduced density matrix to remedy the non-uniqueness and the spurious oscillations in the obtained $v_{xc}(\mathbf{r})$. However, this does not represent the solution of the inverse DFT problem, i.e., the $v_{xc}$ obtained from this approach is not guaranteed to yield the input electron density[23]. Thus, the inverse DFT problem has, heretofore, remained an open challenge.

In this work, we present an advance that provides an accurate solution to the inverse DFT problem, enabling the evaluation of the exact $v_{xc}$ from an ab-initio density. Specifically, the approach uses a finite-element (FE) basis that is systematically convergent and complete, thereby eliminating ill-conditioning in the discrete solution of the inverse DFT problem. Our approach is tested on a range of molecular systems, both weakly and strongly correlated, showing robustness and efficacy in treating realistic polyatomic molecules. The proposed approach therefore unlocks the door to constructing accurate xc functionals that provide precise energies and electronic properties of a huge range of chemical, materials, and biological systems. To elaborate, we envisage the inverse DFT problem to be instrumental in generating $\{\rho^{(i)}, v_{xc}^{(i)}\}$ pairs, using $\rho^{(i)}$'s from correlated ab-initio calculations. Subsequently, these can be used as training data to model $v_{xc}[\rho]$ through machine-learning algorithms[24,25], which are designed to preserve the functional derivative requirement on $v_{xc}[\rho][26]$. Furthermore, the xc energy ($E_{xc}[\rho]$) can be directly evaluated through line integration on $v_{xc}[\rho]$.

## Results

**Constrained optimization for inverse DFT.** We cast the inverse DFT problem of finding the $v_{xc}(\mathbf{r})$ that yields a given density $\rho_{data}(\mathbf{r})$ as the following partial differential equation (PDE)-constrained optimization:

$$\arg \min_{v_{xc}(\mathbf{r})} \int w(\mathbf{r})\big(\rho_{data}(\mathbf{r}) - \rho(\mathbf{r})\big)^2 \, d\mathbf{r}, \quad (1)$$

subject to

$$\left(-\frac{1}{2}\nabla^2 + v_{ext}(\mathbf{r}) + v_H(\mathbf{r}) + v_{xc}(\mathbf{r})\right)\psi_i = \epsilon_i \psi_i, \quad (2)$$

$$\int |\psi_i(\mathbf{r})|^2 \, d\mathbf{r} = 1 . \quad (3)$$

In the above equation, $w(\mathbf{r})$ is an appropriately chosen weight to expedite convergence, $v_{ext}(\mathbf{r})$ represents the nuclear potential, $v_H(\mathbf{r})$ is the Hartree potential corresponding to $\rho_{data}(\mathbf{r})$, and $\psi_i$ and $\epsilon_i$ denote the Kohn–Sham orbitals and eigenvalues, respectively. For simplicity, we restrict ourselves to only closed-shell systems and, hence, the Kohn–Sham density $\rho(\mathbf{r}) = 2\sum_{i=1}^{N_e/2} |\psi_i(\mathbf{r})|^2$. Equivalently, the above PDE-constrained optimization can be solved by minimizing the following Lagrangian,

$$\mathcal{L}\big(v_{xc}, \{\psi_i\}, \{p_i\}, \{\epsilon_i\}, \{\mu_i\}\big) = \int w(\mathbf{r})\big(\rho_{data}(\mathbf{r}) - \rho(\mathbf{r})\big)^2 \, d\mathbf{r}$$
$$+ \sum_{i=1}^{N_e/2} \int p_i(\mathbf{r})\big(\hat{H} - \epsilon_i\big)\psi_i \, d\mathbf{r} + \sum_{i=1}^{N_e/2} \mu_i\left(\int |\psi_i(\mathbf{r})|^2 \, d\mathbf{r} - 1\right),$$
$$(4)$$

with respect to all its constituent variables—$p_i$, $\mu_i$, $\psi_i$, $\epsilon_i$ and $v_{xc}$. In the above equation, $\hat{H} = -\frac{1}{2}\nabla^2 + v_{ext}(\mathbf{r}) + v_H(\mathbf{r}) + v_{xc}(\mathbf{r})$ is

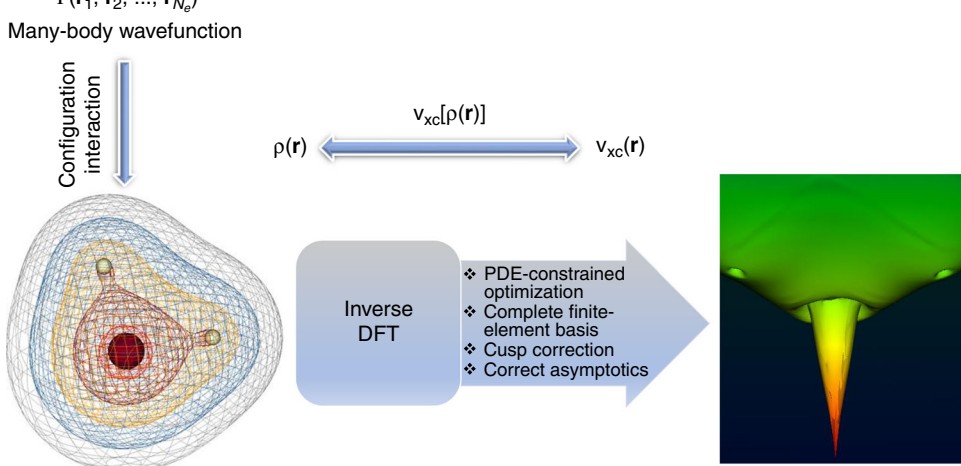

$\Psi(\mathbf{r}_1, \mathbf{r}_2, ..., \mathbf{r}_{N_e})$
Many-body wavefunction

Configuration interaction

$\rho(\mathbf{r})$ ⟷ $v_{xc}[\rho(\mathbf{r})]$ ⟷ $v_{xc}(\mathbf{r})$

Inverse DFT

❖ PDE-constrained optimization
❖ Complete finite-element basis
❖ Cusp correction
❖ Correct asymptotics

**Fig. 1** Schematic of the inverse DFT problem. The exact ground-state many-body wavefunction ($\Psi(\mathbf{r}_1, \mathbf{r}_2, \ldots, \mathbf{r}_{N_e})$) and, hence, the ground-state electron density ($\rho(\mathbf{r})$) is obtained from configuration interaction calculation. The inverse DFT calculation evaluates the exact exchange-correlation potential ($v_{xc}(\mathbf{r})$) that yields the given $\rho(\mathbf{r})$. The ability to accurately solve the inverse DFT problem, presented in this work, presents a powerful tool to construct accurate density functionals ($v_{xc}[\rho(\mathbf{r})]$), either through conventional approaches or via machine learning. The schematic shows the ground-state density and the exact exchange-correlation potential for $H_2O$ obtained in this work

the Kohn–Sham Hamiltonian, $p_i$ is the adjoint function that enforces the Kohn–Sham eigenvalue equation corresponding to $\psi_i$, and $\mu_i$ is the Lagrange multiplier corresponding to the normality condition of $\psi_i$. The optimality of $\mathcal{L}$ with respect to $p_i$, $\mu_i$, $\psi_i$, and $\epsilon_i$ are given by:

$$\hat{H}\psi_i = \epsilon_i \psi_i, \tag{5}$$

$$\int |\psi_i(\mathbf{r})|^2 \, d\mathbf{r} = 1, \tag{6}$$

$$(\hat{H} - \epsilon_i)p_i(\mathbf{r}) = g_i(\mathbf{r}), \tag{7}$$

$$\int p_i(\mathbf{r})\psi_i(\mathbf{r}) \, d\mathbf{r} = 0, \tag{8}$$

where $g_i(\mathbf{r}) = 8w(\mathbf{r})(\rho_{\text{data}}(\mathbf{r}) - \rho(\mathbf{r}))\psi_i - 2\mu_i\psi_i$. We remark that the operator $\hat{H} - \epsilon_i$ in Eq. 7 is singular with $\psi_i$ as its null vector. However, the orthogonality of $g_i$ and $\psi_i$ (consequence of Eq. 7) along with the orthogonality of $p_i$ and $\psi_i$ (Eq. 8) guarantee a unique solution for $p_i$. Having solved the above optimality conditions in Eqs. 5–8, the variation (gradient) of $\mathcal{L}$ with respect to $v_{\text{xc}}$ is given by

$$\frac{\delta\mathcal{L}}{\delta v_{\text{xc}}} = \sum_{i=1}^{N_e/2} p_i \psi_i. \tag{9}$$

This constitutes the central equation for updating $v_{\text{xc}}(\mathbf{r})$ via any gradient-based optimization technique.

Summing up, the proposed approach involves: (i) obtaining $\rho_{\text{data}}(\mathbf{r})$ from correlated ab-initio calculations (i.e., configuration interaction (CI) calculations); (ii) using an initial guess for $v_{\text{xc}}(\mathbf{r})$; (iii) solving Eqs. 5–8 using the current iterate of $v_{\text{xc}}$; (iv) updating $v_{\text{xc}}$ using Eq. 9 as the gradient; (v) repeating (iii)–(iv) until $\rho(\mathbf{r})$ converges to $\rho_{\text{data}}(\mathbf{r})$. We note that the general idea of PDE-constrained optimization has been explored recently in ref. [13]. However, its utility had only been demonstrated on non-interacting model systems in one dimension.

**Verification with LDA-based densities**. To assess the accuracy and robustness of the proposed approach, we use $\rho_{\text{data}}$ obtained from local density approximation (LDA)[27,28]-based DFT calculations, discretized using the FE basis—a systematically improvable and complete basis constructed from piecewise polynomials. This verification test allows us to compare the $v_{\text{xc}}$ obtained from the inverse DFT calculation against $v_{\text{xc}}^{\text{LDA}}[\rho_{\text{data}}]$. As remarked earlier, most of the previous attempts at this verification test have suffered from either non-unique solutions or had resulted in unphysical oscillations in $v_{\text{xc}}$, owing to the incompleteness of the Gaussian basis employed in these works. Figure 2 presents the comparison of $v_{\text{xc}}^{\text{LDA}}[\rho_{\text{data}}]$ against the $v_{\text{xc}}$ obtained from the inverse calculation, for various atomic systems (also see Supplementary Fig. 2). We also provide, in Fig. 3, the $v_{\text{xc}}$ for 1,3-dimethylbenzene ($C_8H_{10}$) obtained from the inverse calculation with LDA-based $\rho_{\text{data}}$ (cf. Supplementary Fig. 3 for the error in $v_{\text{xc}}$), highlighting the efficacy of our approach in accurately treating large systems. We note that all the inverse DFT calculations have been performed in three dimensions and the $L^2$ norm error in the density, $\|\rho_{\text{data}} - \rho\|_{L^2}$, is driven below $10^{-5}$. As evident from these figures, the $v_{\text{xc}}$ determined from the inverse DFT calculation is devoid of any spurious oscillations and is in excellent agreement with $v_{\text{xc}}^{\text{LDA}}[\rho_{\text{data}}]$. In addition, the Kohn–Sham eigenvalues computed using the inverted $v_{\text{xc}}$ are in excellent agreement (i.e., $|\epsilon_i^{\text{LDA}} - \epsilon_i| < 1$ mHa), further validating the accuracy of the method. Although we have reported the verification of our method for LDA-based densities, similar accuracy was obtained using generalized gradient approximation (GGA)-based

densities. We refer to the Supplementary Discussion for a comparison of these verification results against similar studies conducted using existing methods.

**Removing Gaussian basis-set artifacts**. We next turn to employing the proposed method with input densities generated from CI calculations. All the CI calculations reported in this work are performed using the incremental full-CI approach presented in ref. [29] and discretized using the universal Gaussian basis set (UGBS)[30] or polarized triple zeta (cc-PVTZ) Gaussian basis set[31]. It is known that Gaussian basis-set densities, owing to their lack of cusp at the nuclei as well as incorrect far-field decay, induce highly unphysical features in the $v_{\text{xc}}$s obtained from inverse calculations. To this end, we provide two numerical strategies, which, for all practical purposes, remedy the Gaussian basis-set artifacts and thereby allow for accurate evaluation of the exact $v_{\text{xc}}$s from CI densities. It is to be noted that the following numerical strategies are only necessitated due to the unphysical asymptotics in the Gaussian basis-set densities and not due to any inadequacy of the proposed inverse DFT algorithm.

To begin with, the CI density obtained from a Gaussian basis has wrong decay characteristics away from the nuclei (i.e., Gaussian decay instead of exponential decay). This, in turn, results in incorrect asymptotics in the $v_{\text{xc}}$ obtained from an inverse DFT calculation. Thus, to ensure the correct asymptotics in $v_{\text{xc}}$, we employ the following approach. First, we use an initial guess for $v_{\text{xc}}$ that satisfies the correct $-1/r$ decay. In particular, we use the Fermi–Amaldi potential ($v_{\text{FA}}$)[32]. Next, we enforce homogeneous Dirichlet boundary condition on the adjoint function ($p_i$) in the low-density region (i.e., $\rho_{\text{data}} < 10^{-6}$), while solving Eq. 7. In effect, this fixes the $v_{\text{xc}}$ to its initial value in the low-density region, thereby ensuring correct far-field asymptotics in the $v_{\text{xc}}$. This approach is also crucial to obtaining an agreement between the highest occupied Kohn–Sham eigenvalue ($\epsilon_H$) and the negative of the ionization potential ($I_p$), as mandated by the Koopmans' theorem[33,34].

Furthermore, the Gaussian basis-set-based CI densities lack the cusp at the nuclei, which, in turn, leads to undesirable oscillations in the $v_{\text{xc}}$ near the nuclei in any inverse DFT calculation[35–37]. We demonstrate this in the case of equilibrium $H_2$ molecule (bond-length $R_{\text{H−H}} = 1.4$ a.u.), henceforth denoted as $H_2(eq)$. Figure 4 shows the $v_{\text{xc}}$ profile for $H_2(eq)$ corresponding to the $\rho_{\text{data}}(\mathbf{r})$ obtained from a CI calculation, discretized using UGBS. As evident, we observe large unphysical oscillations in the $v_{\text{xc}}$ near the nuclei. We remedy these oscillations by adding a small correction, $\Delta\rho(\mathbf{r})$ to $\rho_{\text{data}}(\mathbf{r})$, so as to correct for the missing cusp at the nuclei. The $\Delta\rho(\mathbf{r})$ is given by

$$\Delta\rho(\mathbf{r}) = \rho_{\text{FE}}^{\text{DFT}}(\mathbf{r}) - \rho_{\text{G}}^{\text{DFT}}(\mathbf{r}), \tag{10}$$

where $\rho_{\text{FE}}^{\text{DFT}}(\mathbf{r})$ is the ground-state density obtained from a forward DFT calculation using a known xc functional (e.g., LDA and GGA) and discretized using the FE basis, and $\rho_{\text{G}}^{\text{DFT}}(\mathbf{r})$ denotes the same, albeit obtained using the Gaussian basis employed in the CI calculation. The key idea here is that $\rho_{\text{FE}}^{\text{DFT}}(\mathbf{r})$, obtained from the FE basis, contains the cusp. Thus, one can expect $\Delta\rho$ to reasonably capture the Gaussian basis-set error near the nuclei. In addition, $\int \Delta\rho(\mathbf{r}) \, d\mathbf{r} = 0$, preserving the number of electrons. A conceptually similar approach has been explored in ref. [37], wherein one post-processes the $v_{\text{xc}}$ instead of pre-processing the $\rho_{\text{data}}$, to remove the oscillations arising from the lack of cusp in $\rho_{\text{data}}$. We illustrate the efficacy of the $\Delta\rho$ correction with the $H_2(eq)$ molecule as an example. Figure 5 presents the $v_{\text{xc}}$ corresponding to the cusp-corrected density (i.e., $\rho_{\text{data}} + \Delta\rho$) for $H_2(eq)$, with two different $\Delta\rho$: $\Delta\rho_{\text{LDA}}$ evaluated using an LDA functional[27,28] and $\Delta\rho_{\text{GGA}}$ evaluated using a GGA functional[38]. As evident, both $\Delta\rho_{\text{LDA}}$- and $\Delta\rho_{GGA}$-based cusp correction generate smooth $v_{\text{xc}}$ profiles. More

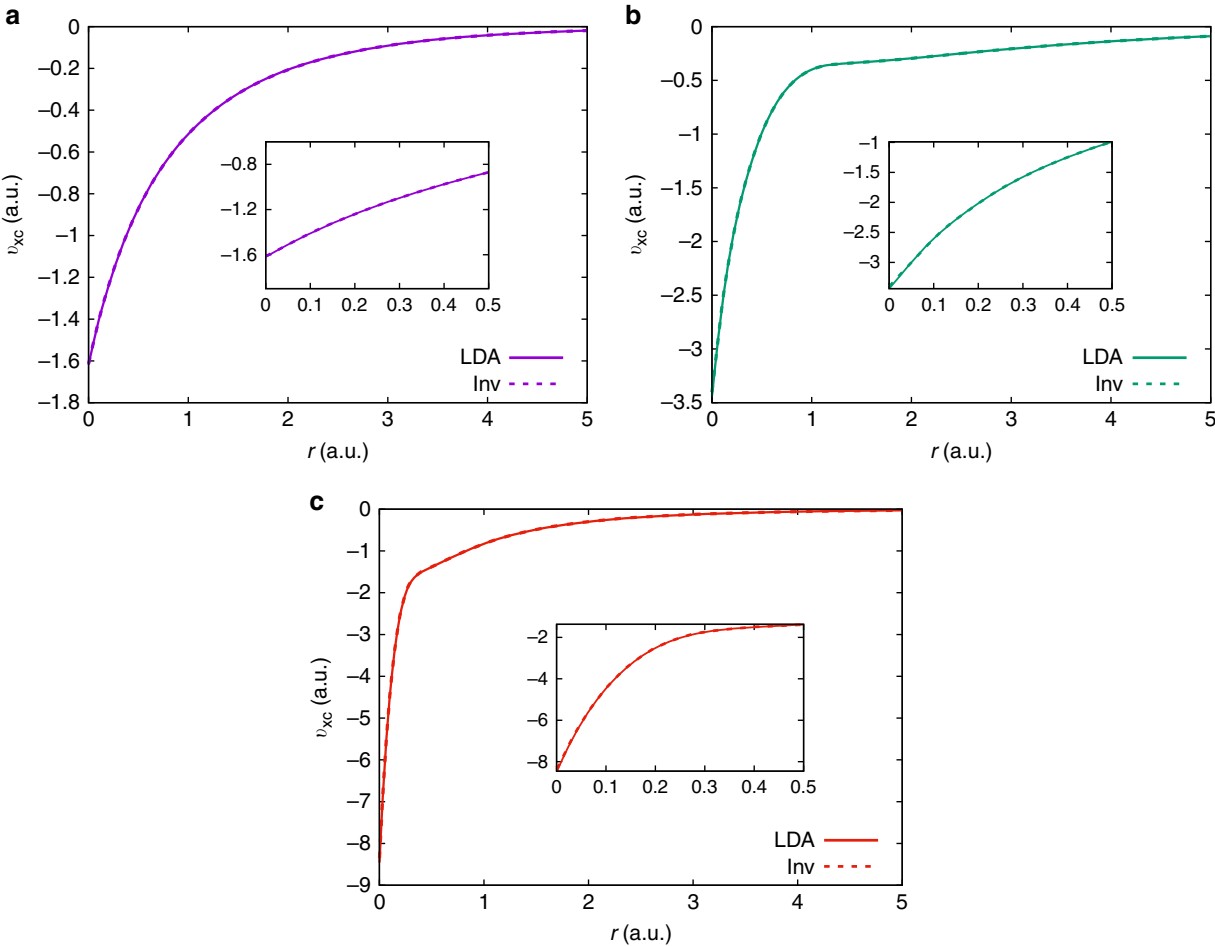

**Fig. 2** Verification study on atomic systems using LDA-based density. The density ($\rho_{data}$) is obtained from a ground-state DFT calculation using an LDA functional. The solid line corresponds to the direct evaluation of the LDA exchange-correlation potential corresponding to $\rho_{data}$, i.e., $v_{xc}^{LDA}[\rho_{data}]$. The dashed line corresponds to the exchange-correlation potential obtained from the inverse DFT calculation using $\rho_{data}$ as the input. The atomic systems considered are as follows: (**a**) He; (**b**) Be; (**c**) Ne

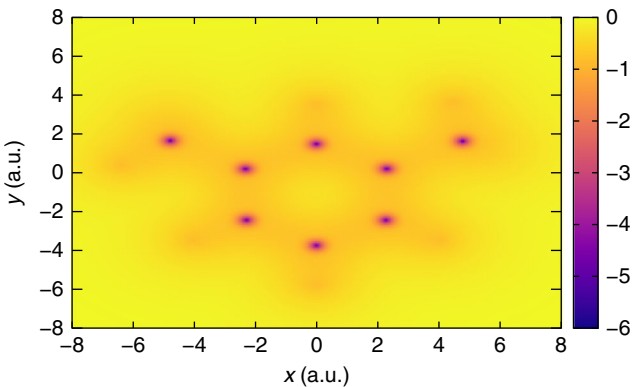

**Fig. 3** Inverse DFT calculation on $C_8H_{10}$. The exchange-correlation potential (in a.u.) determined from the inverse DFT algorithm, using an LDA-based density, is displayed on the plane of the benzene ring. Refer to Supplementary Table 3 for the coordinates

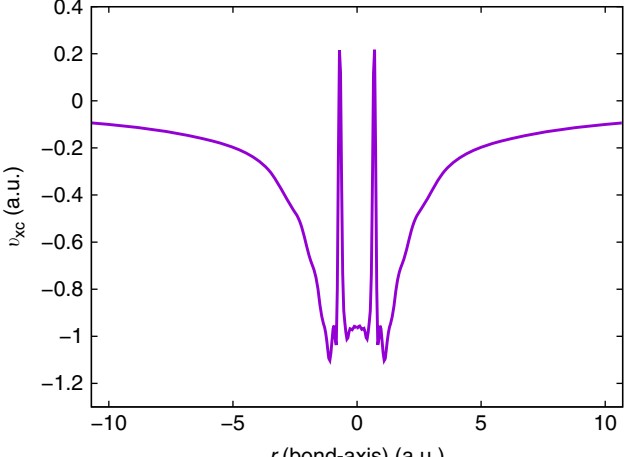

**Fig. 4** Artifact of Gaussian basis-set-based density. The exchange-correlation potential ($v_{xc}$) is evaluated from inverse DFT, using $\rho_{data}$ obtained from a Gaussian basis-set-based configuration interaction (CI) calculation for the equilibrium hydrogen molecule ($H_2(eq)$). The lack of cusp in $\rho_{data}$ at the nuclei induces wild oscillations in the $v_{xc}$ obtained through inversion. The two atoms are located at $r = \pm 0.7$ a.u

importantly, both the profiles are nearly identical, except for small differences in the bonding region between the H atoms. Further, a comparison of both these $v_{xc}$s against the LDA-based xc potential ($v_{xc}^{LDA}$) elucidates the significant difference between the exact $v_{xc}$

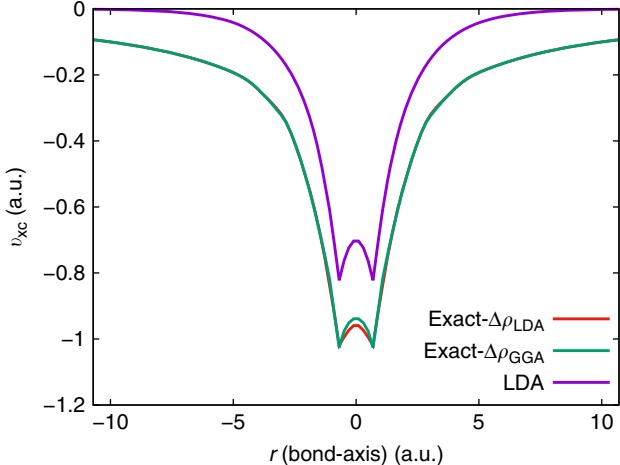

**Fig. 5** Exchange-correlation potentials ($v_{xc}$) for equilibrium $H_2$. A comparison is provided between the exact and the LDA-based $v_{xc}$ potential. The exact exchange-correlation potential is evaluated using the cusp-corrected configuration interaction (CI) density. The effect of the choice of the functional used in evaluating the cusp correction is demonstrated using two different functionals—LDA (exact $- \Delta\rho_{LDA}$) and GGA (exact $- \Delta\rho_{GGA}$)

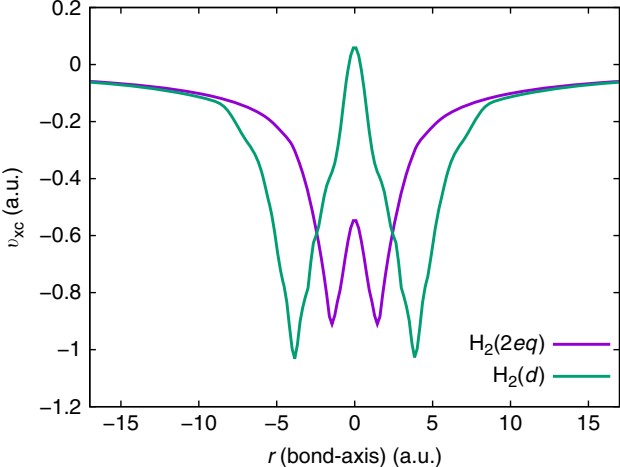

**Fig. 6** Exact $v_{xc}$ for stretched $H_2$ molecules. The exact $v_{xc}$ is provided for two stretched hydrogen molecules: one at twice the equilibrium bond length ($H_2(2eq)$) and the other at dissociation ($H_2(d)$). The H atoms for $H_2(2eq)$ and $H_2(d)$ are located at $r = \pm 1.415$ a.u. and $r = \pm 3.78$ a.u., respectively

**Table 1 Comparison of the highest occupied Kohn–Sham eigenvalue ($\epsilon_H$) and the negative of the ionization potential ($I_p$) (all in Ha)**

|  | $H_2(eq)$ | $H_2(2eq)$ | $H_2(d)$ | $H_2O$ | $C_6H_4$ |
|---|---|---|---|---|---|
| $\epsilon_H$ | −0.601 | −0.482 | −0.479 | −0.452 | −0.354 |
| $-I_p$ | −0.604 | −0.484 | −0.498 | −0.454 | −0.355 |

and $v_{xc}^{LDA}$ even for a simple system that is not strongly correlated. Lastly, for both the $v_{xc}$s, we obtain the same $\epsilon_H$ of $-0.601$ Ha, which, in turn, is in excellent agreement with the $-I_p$ (listed in Table 1). We remark that the agreement of $\epsilon_H$ with $-I_p$ is a stringent test of the accuracy of the inversion and is particularly sensitive to the $v_{xc}$ in the far field.

**Exact $v_{xc}$ from CI densities for molecules**. We now combine the above numerical strategies to evaluate the exact $v_{xc}$ for four other benchmark systems—two stretched $H_2$ molecules and two polyatomic systems (water and ortho-benzyne molecules). The CI calculations for all the molecules, excepting ortho-benzyne, are performed using the UGBS. For ortho-benzyne, we used the cc-PVTZ basis. Given the weak sensitivity of the inverted $v_{xc}$ to the choice of xc functional used in $\Delta\rho$, we employ $\Delta\rho_{LDA}$ for performing the cusp correction in all our calculations. Further, for all the systems, the inverse problem is deemed to have converged when $||\rho_{data} - \rho||_{L^2} < 10^{-4}$. We remark that the $L^2$ error norm is a natural convergence criterion, given the form of the objective function in Eq. 1. However, given that previous works on this inverse problem have reported the $L^1$ error, we provide the same in the Supplementary Table 2, for all the benchmark systems considered. Figure 6 compares the $v_{xc}$ for two stretched $H_2$ molecules—$H_2(2eq)$ ($R_{H-H} = 2.83$ a.u., roughly twice the equilibrium bond length) and $H_2(d)$ ($R_{H-H} = 7.56$ a.u., at dissociation). We emphasize that these are prototypical systems where all existing xc approximations perform poorly, owing to their failure in handling strong correlations. We could successfully solve the inverse DFT problem for these systems ($||\rho_{data} - \rho||_{L^2} \sim 8 \times 10^{-5}$), thereby suggesting that our

approach works equally well for strongly correlated systems. As indicated in Table 1, we get remarkable agreement between $\epsilon_H$ and $-I_p$ for $H_2(2eq)$. However, for $H_2(d)$, we obtain $\epsilon_H$ within 19 mHa of $-I_p$. We attribute this larger difference between $\epsilon_H$ and $-I_p$ (as compared with $H_2(eq)$ and $H_2(2eq)$) to the use of $v_{FA}$ as the boundary condition for $v_{xc}$ in the low-density region. To elaborate, for a single-orbital system, $v_{FA}$ is the exact $v_x$ (exchange-only potential) and, hence, represents the exact $v_{xc}$ in regions where the correlations are negligible. Although for the $H_2(eq)$ and $H_2(2eq)$ molecules the correlations are short-ranged, they are relatively longer-ranged for $H_2(d)$. We highlight this in Fig. 7 by comparing the $v_{xc}$ against $v_x$ for $H_2(eq)$, $H_2(2eq)$, and $H_2(d)$. As evident, $H_2(d)$ has strong correlations extending to a significantly larger domain (in the far-field) in comparison with $H_2(eq)$ and $H_2(2eq)$. Thus, for $H_2(d)$, the use of $v_{FA}$ is warranted only in regions of much lower density (i.e., $\rho_{data} \ll 10^{-6}$) than considered here. However, at such low densities, the wrong far-field asymptotics of a Gaussian basis-set density produces spurious oscillations in the far-field $v_{xc}$. Thus, for the want of more accurate densities, we are restricted to using $v_{FA}$ in regions where $\rho_{data} < 10^{-6}$, at the cost of incurring some error in $\epsilon_H$.

We now turn to a polyatomic system—the $H_2O$ molecule. Figure 8 compares the exact $v_{xc}$ against $v_{xc}^{LDA}$, on the plane of the $H_2O$ molecule. In particular, Fig. 8c provides the comparison along the O–H bond. For the exact $v_{xc}$, we observe an atomic inter-shell structure—marked by a yellow ring around the O atom in Fig. 8b (as well as the local maxima and minima at around $r = \pm 0.4$ a.u. in Fig. 8c). This atomic inter-shell structure is a distinctive feature of the exact $v_{xc}$[39,40] and is absent in the standard xc approximations, as evident from $v_{xc}^{LDA}$. Further, we observe a deeper potential around the O atom, as compared with $v_{xc}^{LDA}$, thereby suggesting a higher electronegativity on the O atom than that predicted by LDA. Moreover, we observe a distinct local maximum at the H atom, as opposed to a local minimum in $v_{xc}^{LDA}$. Lastly, as indicated in Table 1, we obtain striking agreement between $\epsilon_H$ and $-I_p$ for this polyatomic system.

Finally, we evaluate the exact $v_{xc}$ for the singlet state of the ortho-benzyne radical ($C_6H_4$)—a strongly correlated species that has previously served as a test for accurate wavefunction theories[41]. Figure 9 compares the exact $v_{xc}$ against $v_{xc}^{LDA}$, on the plane of the benzyne molecule. This example underscores the

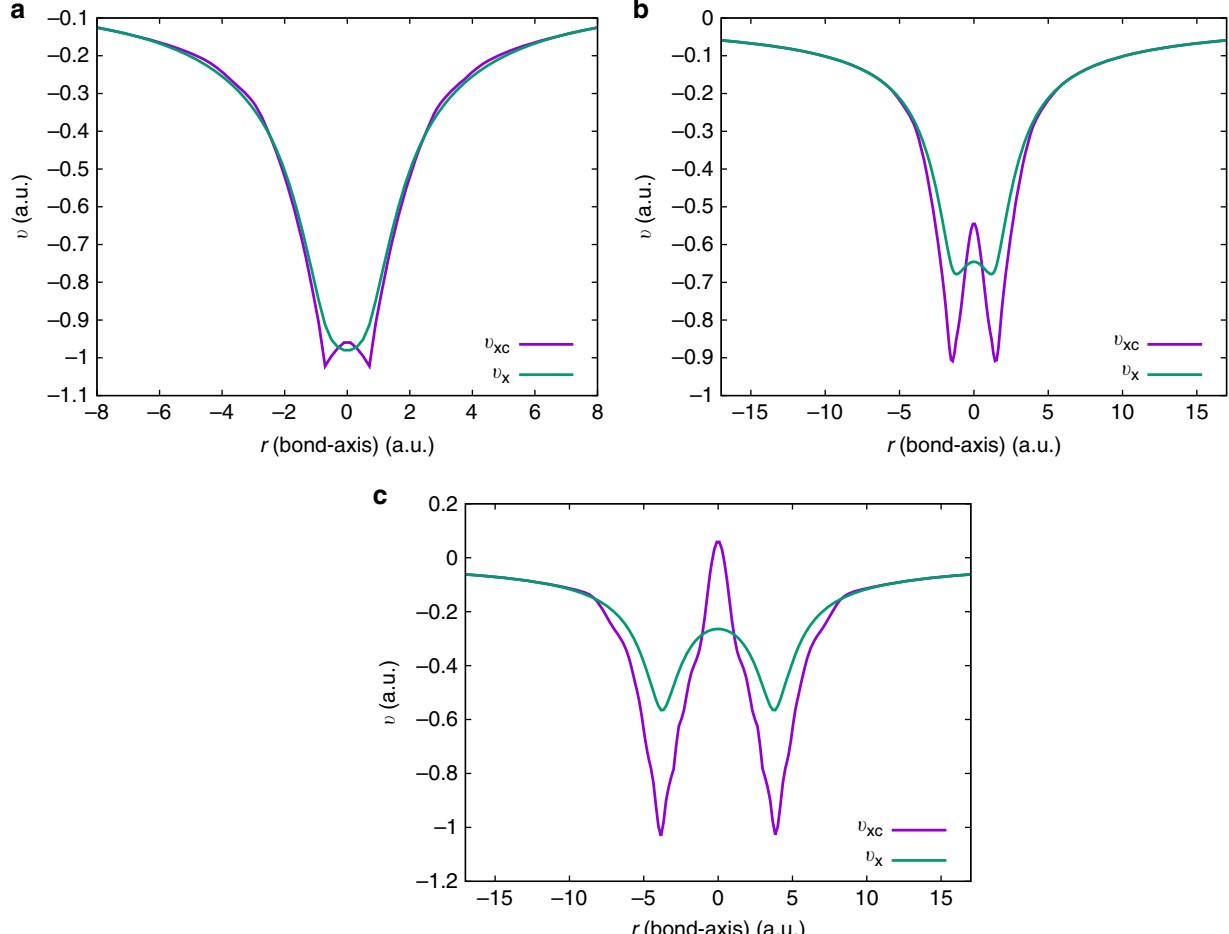

**Fig. 7** Nature and extent of electronic correlations in $H_2$ molecules. A comparison of the exact exchange-correlation ($v_{xc}$) and the exchange-only ($v_x$) potentials is provided for $H_2$ molecules at three different bond lengths: (**a**) equilibrium bond length ($H_2(eq)$); (**b**) twice the equilibrium bond length ($H_2(2eq)$); (**c**) at dissociation ($H_2(d)$). The relative difference between $v_{xc}$ and $v_x$ indicates the nature and extent of electronic correlations. The correlations become stronger with bond stretching

efficacy of our approach in handling both large and strongly correlated systems. As expected for the exact $v_{xc}$, we observe an atomic inter-shell structure—marked by a yellow ring around the C atoms, which, on the other hand, are absent in the case of $v_{xc}^{LDA}$. As is the case with $H_2O$, we observe a deeper potential around the C atom, as compared with $v_{xc}^{LDA}$, suggesting a higher electronegativity on the C atom than that predicted by LDA. Furthermore, as indicated in Table 1, we obtain remarkable agreement between $\epsilon_H$ and $-I_p$.

## Discussion

We have presented an accurate and robust method to evaluate the exact $v_{xc}$, solely from the ground-state electron density. The key ingredients in our approach are—(a) the effective use of FE basis, which is a systematically convergent and complete basis, and, in turn, results in a well-posed inverse DFT problem; (b) the use of $\Delta\rho$ correction and appropriate far-field boundary conditions to alleviate the unphysical artifacts associated with Gaussian basis-set densities. We emphasize that the proposed approach can easily drive the error in the target densities, i.e., $||\rho_{data} - \rho||_{L^2}$, to tight tolerances of $\mathcal{O}(10^{-5} - 10^{-4})$—which represents a stringent accuracy (see the Supplementary Discussion for a comparison with existing methods). Notably, as demonstrated through the 1,3-dimethylbenzene and the orthobenzyne molecules, our approach can competently handle

system sizes, which have, heretofore, remained challenging for other inverse DFT methods. Furthermore, for all the exact $v_{xc}$s obtained from CI densities, we obtain excellent agreement between $\epsilon_H$ and $-I_p$ (excepting in the case of $H_2(d)$), further validating the accuracy and robustness of the approach. We remark that the larger discrepancy between $\epsilon_H$ and $-I_p$ in the case of $H_2(d)$ is a consequence of long-range (static) correlations in this system coupled with incorrect far-field asymptotics of Gaussian basis-set densities and can be remedied with the availability of more accurate densities. The ability to evaluate the exact xc potentials from ground-state electron densities, enabled by this method, will provide a powerful tool in the future testing and development of approximate xc functionals. Further, it paves the way for using machine learning to construct the functional dependence of $v_{xc}$ on $\rho$, i.e., $v_{xc}[\rho]$, providing another avenue to develop density functionals[24,42,43] that can systematically improve both ground-state densities and energies[44] as well as satisfy the known conditions on the exact functional[45–47].

## Methods

**Discretization**. We employ spectral FE basis to discretize all the spatial fields—$v_{xc}$, $\{\psi_i\}$, $\{p_i\}$. The FE basis is constructed from piecewise polynomials on non-overlapping subdomains called elements. The basis, thus constructed, can be systematically improved to completeness by reducing the element size and/or increasing the polynomial order[48]. We remark that the spectral FE basis are not

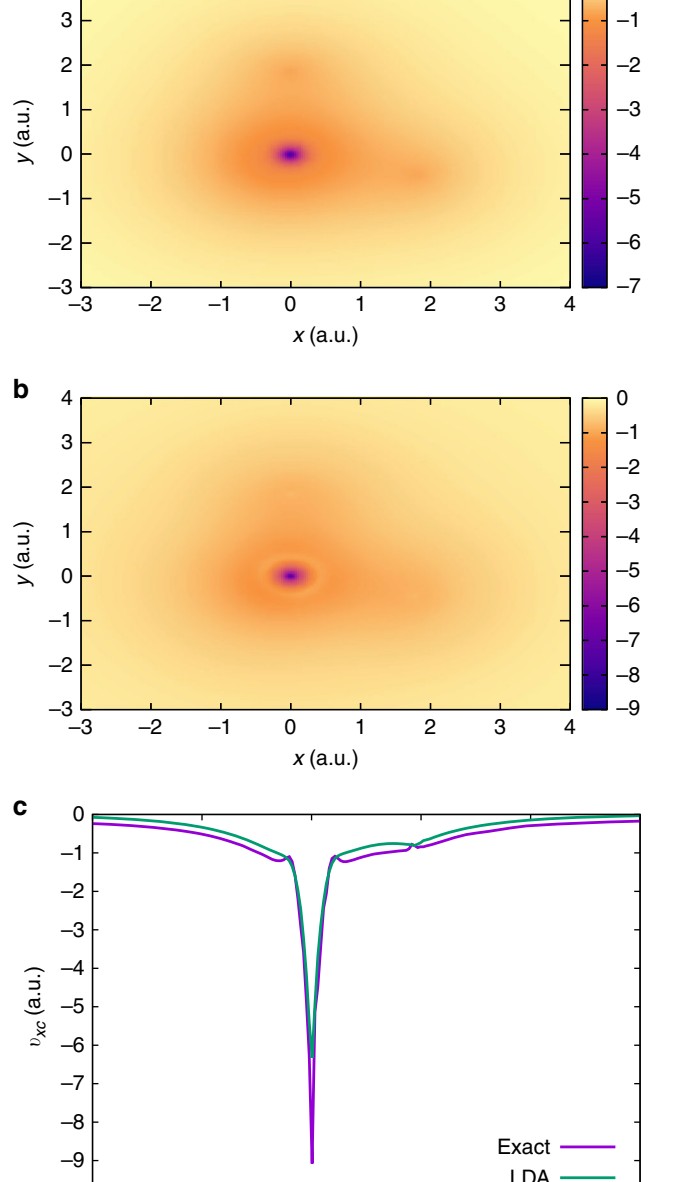

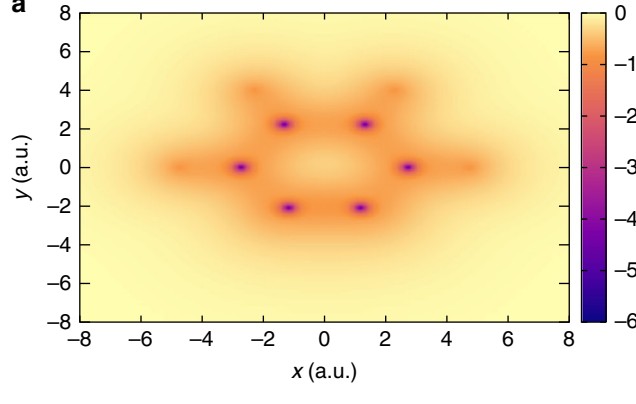

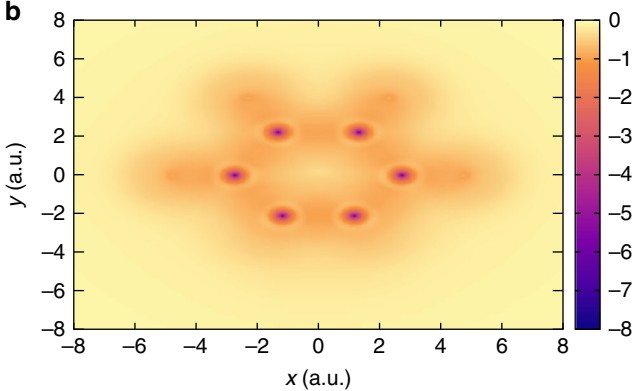

**Fig. 9** Comparison of exchange-correlation potentials ($v_{xc}$) for $C_6H_4$. **a** LDA-based exchange-correlation potential. **b** Exact exchange-correlation potential. In both the cases, the $v_{xc}$ (in a.u.) is presented on the plane of the molecule. Refer to Supplementary Table 3 for the coordinates

**Fig. 8** Comparison of exchange-correlation potentials ($v_{xc}$) for $H_2O$. **a** LDA-based exchange-correlation potential. **b** Exact exchange-correlation potential. **c** Comparison of the LDA-based and the exact exchange-correlation potential along the O–H bond. In **a** and **b**, the $v_{xc}$ (in a.u.) is presented on the plane of the molecule. Refer to Supplementary Table 3 for the coordinates

orthogonal and, hence, result in a generalized eigenvalue problem as opposed to the more desirable case of standard eigenvalue problem. To this end, we use special reduced-order quadrature (Gauss–Legendre–Lobatto quadrature rule) to render the overlap matrix diagonal and, thereby, trivially transform the generalized eigenvalue problem into a standard one. For all the $H_2$ molecules, we used adaptively refined quadratic FEs to discretize the $\{\psi_i\}$ and $\{p_i\}$, whereas for all other systems we used adaptively refined fourth-order FEs. The $v_{xc}$, in all the calculations, is discretized using linear FEs. Most importantly, the form of the FE basis is chosen carefully, so as to guarantee the cusp in $\psi_i$s (and hence in $\rho$) at the nuclei, which in turn is critical to obtaining accurate $v_{xc}$s near the nuclei (refer to the Supplementary Note 1 for more information).

**Solvers**. In order to efficiently solve the Kohn–Sham eigenvalue problem in Eq. 5, we employ the Chebyshev polynomial-based filtering technique[48–50]. We remark that, compared with a forward ground-state DFT calculation, the inverse DFT calculation warrants much tighter accuracy in solving the Kohn–Sham eigenvalue equation(s). However, the use of a very high polynomial degree Chebyshev filter can generate an ill-conditioned subspace, akin to any power iteration-based eigen solver. To circumvent the ill conditioning and attain higher accuracy, we employ multiple passes of a low polynomial degree Chebyshev filter (polynomial order ~ 1000) and orthonormalize the Chebyshev-filtered vectors between two successive passes. The number of passes is determined adaptively so as to guarantee an accuracy of $10^{-9}$ in $||\hat{H}\psi_i - \epsilon_i\psi_i||_{L^2}$.

The discrete adjoint function ($p_i$) is solved by, first, projecting Eq. 7 onto a space orthogonal to the corresponding $\psi_i$ (or degenerate $\psi_i$s) and then employing the conjugate-gradient method to compute the solution. The discrete adjoint problem is solved to an accuracy of $10^{-12}$ in $||(\hat{H} - \epsilon_i)p_i - g_i||_{L^2}$.

The update for $v_{xc}$ is computed using limited-memory Broyden-Fletcher-Goldfarb-Shanno (BFGS) algorithm, a memory-efficient quasi-Newton solver, which constructs approximate Hessian matrices using the history of the gradients[51]. In all the calculations, we used a history of size 100 to construct the approximate Hessian. We refer to Supplementary Discussion for details on the rate of convergence and the factors affecting it.

**Weights**. To expedite the convergence of the nonlinear solver, we make use of two different weights, $w(\mathbf{r}) = 1$ and $w(\mathbf{r}) = 1/\rho_{data}^{\alpha}$ ($1 \leq \alpha \leq 2$), in sequence. The latter penalizes the objective function in the low-density region.

**Ab initio densities**. Accurate electron densities were generated using the incremental full CI (iFCI) method[29] in the Q-Chem software package[52]. This method solves the electronic Schrödinger equation via a many-body expansion and asymptotically produces the exact electronic energy and density as the number of bodies in the expansion approaches the all-electron limit. For this study, electron densities were provided in the all-valence-electron limit of iFCI, i.e., where the full valence set is fully correlated and the core orbitals of $H_2O$ and $C_6H_4$ are treated as uncorrelated electron pairs. Reference ionization energies were obtained at the same level of theory, for each system with one less electron.

## Data availability

The authors declare that all the data supporting the results of this study are available upon reasonable request to the corresponding author.

## Code availability

The code used to perform inverse DFT calculations is available upon reasonable request to the corresponding author.

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

## Acknowledgements

We thank O. Ghattas for fruitful discussions. We are grateful for the support of Toyota Research Institute under the auspices of which this work was conducted. We also acknowledge the support of Department of Energy, Office of Science, under grant number DE-SC0017380, which supported the development of all-electron DFT calculations using the finite-element basis that was instrumental in this work. V.G. also acknowledges the

support of Air Force Office of Scientific Research under grant number FA-9550-17-1-0172, which supported the development of some aspects of the computational framework used in this work. This research used resources of the National Energy Research Scientific Computing Center, a DOE Office of Science User Facility supported by the Office of Science of the US Department of Energy under Contract Number DE-AC02-05CH11231. This work used the Extreme Science and Engineering Discovery Environment (XSEDE), which is supported by National Science Foundation Grant number ACI-1053575. V.G. also gratefully acknowledges the support of the Army Research Office through the DURIP grant W911NF1810242, which also provided computational resources for this work.

## Author contributions

V.G. designed the research. B.K. performed the inverse DFT calculations. P.M.Z. performed the CI calculations. B.K. and V.G. analyzed the data. All authors contributed to the manuscript.

## Competing interests

The authors declare no competing interests.
