## [Peer Review File · Nature Communications]

Reviewers' comments:

Reviewer #1 (Remarks to the Author):

Solving the inverse DFT problem, i.e. obtaining (numerically) exact exchange-correlation potentials from electron densities, is one of the most fundamental and at the same time numerically challenging problems in density functional theory. As a consequence, it has attracted great research interest over the past decades. While several important steps forward have been made, essentially all approaches in use so far suffer from numerical issues, which prevent a robust and systematically convergent solution of this problem.

The great value and novelty of the present work lies in the fact that it properly and simultaneously addresses all the numerical challenges preventing an accurate inverse solution so far. This includes, in particular, (i) input electron densities of very high quality (incremental Full Configuration Interaction, iFCI), (ii) a formulation of the potential evaluation as a PDE-constrained optimization problem, which (iii) is solved in a finite-element basis that can be systematically converged to completeness, (iv) an initial guess for the XC potential with correct asymptotic behavior, which is preserved in the calculation through boundary conditions, and (v) a cusp-correction for the input electron densities, which is necessary since the iFCI densities are represented in a Gaussian basis set.

Examples include atomic systems, the H₂ molecule at different distances, and H₂O as a simple polyatomic molecule. Though only small molecules are considered, this set of examples is convincing and well-suited to demonstrate the characteristics and effectiveness of the presented approach.

Specific comments:

(1) The example of H₂(d), i.e. the H₂ molecule at dissociation, highlights that long-range (static) correlation may pose a challenge for the present approach, as the correct long-range behavior of the potential is only ensured if correlation contributions decay fast enough. A short discussion of the implications and possible limitations arising from this observation could be included in the Discussion section.

(2) The introduction implies that this approach "unlocks the door to ... highly accurate exchange-correlation functionals that provide precise energies". However, the road from potentials for specific densities to general density functionals is not obvious, and the discussion remains rather general and vague about this point. Can the authors briefly elaborate on the question of how their approach can specifically help to obtain accurate energy functionals from the potentials? Or should the statement in the introduction be adapted?

(3) The important aspect of the choice of the FE basis remains somewhat unclear. Can the authors add specific information, in particular on how the choice of the FE basis guarantees the correct cusp behavior?

(4) How were the reference ionization potentials obtained (Table 1)?

Reviewer #2 (Remarks to the Author):

The paper claims that it finally solves the long-standing problem of mapping a given ground-state electron density to its exchange-correlation potential. This claim is greatly exaggerated for the following reasons.

1. Except for Gaussian basis-set densities, construction of exchange-correlation (xc) potentials

from electron densities for small systems does not pose much difficulty. If the input density is generated using Slater-type basis functions or a very dense numerical grid, then the methods of Refs. 12, 20, and 32 would produce at least as good results as the proposed method in Fig. 2 (which, by the way, does not show in detail what happens at very small and very large r).

2. For the same accurate potential, the L2 norm density error used in this work is orders of magnitude smaller than the L1 norm used in other methods such as that of Ref. 32. The convergence thresholds of $10^{**(-5)}e^{**2}$ and $10^{**(-6)}e^{**2}$ used in the paper are probably equivalent to the L1 norm error of $\sim 0.01e$, which is much greater than the values used in Refs. 32 and elsewhere. It is very misleading to claim "unprecedented" L2-norm errors when most readers familiar with the problem are thinking in terms of L1 norms.

3. The authors attempt to use Gaussian basis-set densities to produce xc potentials that do not oscillate and decay as $-1/r$. Such attempts are misguided because Kohn-Sham potentials corresponding to Gaussian densities in the basis-set limit must oscillate and increase as r^{**2} at large r , as shown in Refs. 31 and 32. Any potential from a Gaussian density that does not oscillate (at least near the nucleus) or does not blow up is not the exact answer to the inversion problem.

4. The proposed method gets rid of the oscillations by "adding a small correction, $\Delta \rho(r)$, to $\rho_{\text{data}}(r)$, so as to correct for the missing cusp at the nuclei" (eq. 6). In other words, the "solution" is to fudge the input density. This is hardly "an advance that is likely to influence thinking in the field". Moreover, the principle of this correction is clearly inspired by the trick of Ref. 33, a fact that is not acknowledged.

5. The authors apply the following adjectives to describe their method: "rigorous", "unambiguous", "unprecedented", etc. None of these characterizations is justified. Modifying an input density is neither "rigorous" nor "unambiguous". Convergence with respect to the L2 norm of $10^{**(-6)}e^{**2}$ is anything but "unprecedented", especially after ρ_{data} is "corrected". Wobbly potentials seen in Figs. 5 and 6c are poorer than could be obtained by other methods.

6. The general idea of using constrained optimization for mapping densities to xc potentials is not new. Refs. 12, 13, 18, 20 employ similar principles.

In summary, the major claims of the paper are not novel, the conclusions are not original, and the work is not convincing overall. Therefore, it does not merit publication.

Reviewer #3 (Remarks to the Author):

In "Exact exchange-correlation potentials from ground-state electron densities" the authors present an algorithm to solve the 'inversion' problem in DFT. Where given an electron density, one can obtain the Kohn-Sham potential that will result in the density. This has been an open problem in the field for many years, and an efficient solution to this problem is an important step for a variety of different fields. The paper is well written and clear. Therefore, I believe this work should be considered for publication in Nature Communications; however, I believe there should be several points to address before the manuscript is published.

1. As the authors note, partial differential equation constrained optimization has been explored by other authors before but only in model/test systems in 1-dimension. In many ways, the difficulty in the inverse problem is the numerical solution. The Wu and Yang approach with regularization should provide an 'optimal' solution and in relatively simple systems one can numerically find that solution. However, when one tries to apply it to large systems, then it becomes numerically intractable. Therefore, my largest concern about this work, is that this approach works great for small systems, but is still numerically intractable to any sizeable system. The largest systems presented here are 10 electrons. I think to showcase that this is a significant step forward, they

authors should provide numerical results for a large system (50-100 electrons). If one can still solve the inverse problem using this partial differential equation constrained optimization methodology, it will really show that this work is a breakthrough in this field.

2. As stated above, it is the numerical difficulty which really makes this problem hard. Therefore, how is the convergence of these equations? Specifically, on convergence of $\|\rho_{data} - \rho\|_{L^2}$: (a) How many cycles does the density take to converge? (b) How is the convergence affected by the initial V_{xc} guess? (c) How is the convergence affected by the number of electrons in the system? (d) How is convergence affected by the weak, eg $H_2(eq)$, and strong correlation, eg $H_2(d)$, systems?

3. In the H_2 stretched/dissociated case, what is the orbital energy of the LUMO going to? Are there significant difficulties in solving the strong correlation system with multiple near-degenerate orbitals? Naively, I would have expected that during the optimization cycles you could have potentials, which make the original LUMO become the HOMO as you try to force the strong-correlation solution. Does the optimization process continue to be well behaved?

Again, I want to emphasize that I believe this is important work. However, there have been several inverse methods that have been proposed over the last 15 years that have been applied to small systems that work relatively well. The difficulty in the problem is in the numerics and I think the authors should discuss and show that this optimization scheme is numerically tractable and simple to be able to deal with large and complex systems.

Response to Reviewers

Bikash Kanungo, Paul M. Zimmerman, Vikram Gavini

We thank the reviewers for the careful review of the manuscript, and their comments and suggestions to improve the manuscript. Below we are providing a detailed response to the reviewers' comments and the changes made to the manuscript. We hope we have satisfactorily addressed all the questions and concerns of the reviewers.

Response to Reviewer #1

1. *Solving the inverse DFT problem, i.e. obtaining (numerically) exact exchange-correlation potentials from electron densities, is one of the most fundamental and at the same time numerically challenging problems in density functional theory. As a consequence, it has attracted great research interest over the past decades. While several important steps forward have been made, essentially all approaches in use so far suffer from numerical issues, which prevent a robust and systematically convergent solution of this problem. The great value and novelty of the present work lies in the fact that it properly and simultaneously addresses all the numerical challenges preventing an accurate inverse solution so far. This includes, in particular, (i) input electron densities of very high quality (incremental Full Configuration Interaction, iFCI), (ii) a formulation of the potential evaluation as a PDE-constrained optimization problem, which (iii) is solved in a finite-element basis that can be systematically converged to completeness, (iv) an initial guess for the XC potential with correct asymptotic behavior, which is preserved*

in the calculation through boundary conditions, and (v) a cusp-correction for the input electron densities, which is necessary since the iFCI densities are represented in a Gaussian basis set.

Response: We thank the reviewer for commenting on the fundamental as well as challenging nature of the inverse DFT problem, and for summarizing the value and novelty of this work in solving this problem.

- 2. Examples include atomic systems, the H₂ molecule at different distances, and H₂O as a simple polyatomic molecule. Though only small molecules are considered, this set of examples is convincing and well-suited to demonstrate the characteristics and effectiveness of the presented approach.*

Response: We remark that, in response to comments from reviewer 3, we have included additional examples to demonstrate the efficacy of the proposed approach on larger complex systems. In particular, the revised manuscript includes the results from 1,3-dimethylbenzene molecule and *ortho*-benzyne radical (a strongly correlated system).

- 3. The example of H₂(d), i.e. the H₂ molecule at dissociation, highlights that long-range (static) correlation may pose a challenge for the present approach, as the correct long-range behavior of the potential is only ensured if correlation contributions decay fast enough. A short discussion of the implications and possible limitations arising from this observation could be included in the Discussion section.*

Response: We have added a short discussion on this point in the Discussion section. We remark that this limitation, in the context of systems with long-range static correlation, is a consequence of the wrong far-field decay of Gaussian basis-set densities, and not a shortcoming of the inverse procedure itself. We expect better accuracy for systems with such long-range (static) correlation with the availability of more accurate densities.

4. *The introduction implies that this approach “unlocks the door to ... highly accurate exchange-correlation functionals that provide precise energies”. However, the road from potentials for specific densities to general density functionals is not obvious, and the discussion remains rather general and vague about this point. Can the authors briefly elaborate on the question of how their approach can specifically help to obtain accurate energy functionals from the potentials? Or should the statement in the introduction be adapted?*

Response: We have added a brief discussion following this sentence in the revised manuscript. The path we envision involves generating $\{\rho^{(i)}(\mathbf{r}), v_{\text{xc}}^{(i)}(\mathbf{r})\}$ pairs through the proposed inverse DFT approach, as a first step. Subsequently, we intend to use them as training data for machine-learning the functional dependence of v_{xc} on ρ , i.e., learn $v_{\text{xc}}[\rho]$. This second step will rely on recent developments in function-to-function learning. Recent studies in function-to-function learning in DFT have explored Kernel Ridge Regression [1] or Neural Networks [2]. We intend to extend and improve on these machine-learning techniques to model $v_{\text{xc}}[\rho]$, while at the same time enforcing some of the known exact conditions on $v_{\text{xc}}[\rho]$ [3]. Subsequently, one can perform a line integral on $v_{\text{xc}}[\rho]$ to evaluate $E_{\text{xc}}[\rho]$.

5. *The important aspect of the choice of the FE basis remains somewhat unclear. Can the authors add specific information, in particular on how the choice of the FE basis guarantees the correct cusp behavior?*

Response: We have added additional information on the choice of the FE basis and how it admits the cusp, in the SI submitted with the revised manuscript. Briefly, we use Lagrange polynomials as FE basis functions that are C^0 continuous on the boundary of an element. Furthermore, the FE mesh is adapted such that the nuclei are positioned on the corner nodes of elements, thereby admitting a cusp owing to the C^0 continuity of the basis on the element boundary.

6. *How were the reference ionization potentials obtained (Table 1)?*

Response: The reference ionization potentials were obtained using the same method as for the reference electron densities (incremental full CI). Specifically, one electron was removed from the system and the total energy recomputed, the ionization energy being the difference between the original energy and the minus-one-electron energy. This point has been noted in the methods section.

Response to Reviewer #2

1. *Except for Gaussian basis-set densities, construction of exchange-correlation (xc) potentials from electron densities for small systems does not pose much difficulty. If the input density is generated using Slater-type basis functions or a very dense numerical grid, then the methods of Refs. 12, 20, and 32 would produce at least as good results as the proposed method in Fig. 2 (which, by the way, does not show in detail what happens at very small and very large r).*

Response: **The completeness of the basis—in which the Kohn-Sham orbitals ($\psi_i(\mathbf{r})$) as well as the exchange-correlation potential ($v_{xc}(\mathbf{r})$) are discretized—is of utmost importance in an inverse DFT calculation, irrespective of the basis-set used in generating ρ_{data} .** It has been highlighted, both theoretically and numerically, that the lack of completeness in the basis-set employed in discretizing ψ_i 's and v_{xc} can result in an ill-posed problem, thereby producing non-unique solutions and/or spurious oscillations in v_{xc} [4, 5, 6]. Thus, the use of finite-element basis adopted in this work, which can systematically be converged to completeness, is crucial to resolving the numerical challenges associated with the inverse DFT problem.

While [7, 6] (Refs. 12 and 20 in the Reviewer's comment) use constrained optimization approach to the inverse DFT problem, the lack of a complete basis in discretizing the inverse problem, results in both qualitative and quantitative errors in the inverted v_{xc} . To elaborate: (i) in [7] (Ref. 12 in the Reviewer's comment), the v_{xc} for H_2 (Fig. 3 in [7]) does not have a local maximum between the two H atoms, a feature present in both the exact as well approximate v_{xc} 's (cf. Fig. 5 in our revised manuscript);

(ii) in [6] (Ref. 20 in the Reviewer’s comment), the v_{xc} obtained has an error of $\mathcal{O}(1)$ in the verification test against LDA. Further, it is sensitive to the choice of regularization parameter, thereby suggesting that one cannot dispense with the use of a complete basis, even after introducing regularization to the objective function; (iii) in [8] (Ref. 32 in Reviewer’s comment) the applicability of the linear-response method (and the assumptions therewithin) to molecular systems is yet to be established (all the calculations in [8] are restricted to atomic calculations).

However, we recognize the reviewer’s concern that the range of systems demonstrated in the previous manuscript do not show the full power of our methodology. **To address this concern, we have added two larger molecules to the revised manuscript and confirmed that accurate xc potentials can still be straightforwardly constructed using our method.** The 1,3,-dimethylbenzene (C_8H_{10}) and ortho-benzyne (C_6H_4) molecules have 58 and 40 electrons, respectively, which are unprecedented for inverse DFT calculations. The ease in generating xc potentials for these molecules confirms the advances presented herein.

For the atomic systems considered in our verification studies (cf. Fig. 2), we have also included (in the SI) the comparison of the inverted v_{xc} against $v_{xc}^{LDA}[\rho_{data}]$ in both small and large \mathbf{r} regimes.

2. *For the same accurate potential, the L2 norm density error used in this work is orders of magnitude smaller than the L1 norm used in other methods such as that of Ref. 32. The convergence thresholds of $10^{**}(-5)e^{**2}$ and $10^{**}(-6)e^{**2}$ used in the paper are probably equivalent to the L1 norm error of $\sim 0.01e$, which is much greater than the values used in Refs. 32 and elsewhere. It is very misleading to claim “unprecedented” L2-norm errors when most readers familiar with the problem are thinking in terms of L1 norms.*

Response: The reviewer points to an important aspect of confirming the relevance of this work. **However, the convergence threshold employed in this work (in terms of the L^2 norm: $\|\rho(\mathbf{r}) - \rho_{data}(\mathbf{r})\|_{L^2} = \sqrt{\int (\rho(\mathbf{r}) - \rho_{data}(\mathbf{r}))^2 d\mathbf{r}}$) does not**

imply an L^1 norm of $\sim 0.01e$, as suggested by the reviewer. In fact, in all the calculations, except $H_2(d)$, we obtain L^1 norms of $\mathcal{O}(10^{-5} - 10^{-4})$ (details included in SI submitted with the revised manuscript). We underline that this L^1 error of $\mathcal{O}(10^{-5} - 10^{-4})$ is unprecedented for 3D inverse DFT calculations (while similar accuracies have been obtained in [8] (Ref. 32 in Reviewer’s comment), they are all restricted to 1D calculations.)

3. *The authors attempt to use Gaussian basis-set densities to produce xc potentials that do not oscillate and decay as $-1/r$. Such attempts are misguided because Kohn-Sham potentials corresponding to Gaussian densities in the basis-set limit must oscillate and increase as r^{**2} at large r , as shown in Refs. 31 and 32. Any potential from a Gaussian density that does not oscillate (at least near the nucleus) or does not blow up is not the exact answer to the inversion problem.*

Response: While the v_{xc} corresponding to Gaussian basis-set densities exhibit oscillations and r^2 growth at large r , these are precisely Gaussian-basis set artifacts and not the features of the physical v_{xc} . The focus of the present work is not to reproduce the Gaussian basis-set artifacts (which we demonstrate in Figure 4 of the revised manuscript for H_2), but rather to use Gaussian basis-set densities (as they are efficient for CI calculations) and still be able to evaluate the physically exact v_{xc} . To this end, our approach of using $\Delta\rho$ cusp-correction, as well as the far-field boundary condition ensuring correct asymptotics for v_{xc} , provide a robust means to correct for the Gaussian basis-set artifacts and compute the physically exact v_{xc} .

4. *The proposed method gets rid of the oscillations by “adding a small correction, $\Delta\rho$, to ρ_{data} , so as to correct for the missing cusp at the nuclei” (eq. 6). In other words, the “solution” is to fudge the input density. This is hardly “an advance that is likely to influence thinking in the field”. Moreover, the principle of this correction is clearly inspired by the trick of Ref. 33, a fact that is not acknowledged.*

Response: We note that the $\Delta\rho$ -correction ‘corrects’ for the missing cusp in the

Gaussian input density, and is far from being an ad-hoc fudging of the input density. We hypothesize that the $\Delta\rho$ correction encapsulates the Gaussian basis-set error near the nuclei. Evidence to this hypothesis is provided in Fig. 5 of the revised manuscript, wherein we observe only weak sensitivity of the inverted v_{xc} to the xc functional used in constructing $\Delta\rho$. This, in our view, constitutes a robust numerical strategy to mitigate the wild oscillations induced by the missing cusp at the nuclei in Gaussian densities.

We agree that there is similarity of our $\Delta\rho$ correction to that of Δv_{osc} in [9]. We have acknowledged this on page 8 of the revised manuscript.

5. *The authors apply the following adjectives to describe their method: “rigorous”, “unambiguous”, “unprecedented”, etc. None of these characterizations is justified. Modifying an input density is neither “rigorous” nor “unambiguous”. Convergence with respect to the L_2 norm of $10^{-6}e^2$ is anything but “unprecedented”, especially after ρ_{data} is “corrected”. Wobbly potentials seen in Figs. 5 and 6c are poorer than could be obtained by other methods.*

Response: We first note that we did not use “rigorous” to describe our work. We refer to the responses in #2,#3 and #4 for more detailed justification for the ‘unambiguous’ solution to the inverse DFT problem made possible by this work, and ‘unprecedented’ accuracy obtained. Briefly, we qualify our approach as both “robust” and “unambiguous” owing to the stability of the algorithm, non-sensitivity to the initial guess, and above all the uniqueness of the converged v_{xc} . Furthermore, we qualify our work as “unprecedented” on the basis of attaining L^2 and L^1 errors of $\mathcal{O}(10^{-5} - 10^{-4})$ in the density for 3D problems, as well as having an excellent agreement between ϵ_H and $-I_p$.

We note that the slight wobbly nature of the v_{xc} only appears in the case of the dissociated hydrogen molecule ($H_2(d)$), and not in other molecules. We emphasize that $H_2(d)$ is a particularly difficult system for inverse DFT, and this work is the first to attempt to accurately evaluate the exact v_{xc} for a dissociated molecule. We suspect the slight wobbliness to be a consequence of the Gaussian basis-set density. However, given the difficulty in solving an inverse problem for a dissociated system, we consider

the quality of our v_{xc} to be quite good.

6. *The general idea of using constrained optimization for mapping densities to xc potentials is not new. Refs. 12, 13, 18, 20 employ similar principles.*

Response: We agree that the idea of constrained optimization, in the context of inverse DFT, has been explored in the past, and we do not claim the PDE-constrained optimization in itself as the novelty of this work. As noted by Reviewer 1, the novelty of the work lies in combining the following to solve the outstanding inverse DFT problem: PDE-constrained optimization, use of finite-elements that can be systematically converged to completeness, the cusp-correction in Gaussian densities, and use of far-field boundary conditions that ensures the correct asymptotic behavior for v_{xc} . **In total, the proposed technique provides the ability to invert xc potentials for molecules that are far larger than ever before (40 or more electrons), demonstrating a systematic means to determine v_{xc} for systems well beyond the scope of prior methods.**

Response to Reviewer #3

1. *In “Exact exchange-correlation potentials from ground-state electron densities” the authors present an algorithm to solve the ‘inversion’ problem in DFT. Where given an electron density, one can obtain the Kohn-Sham potential that will result in the density. This has been an open problem in the field for many years, and an efficient solution to this problem is an important step for a variety of different fields. The paper is well written and clear. Therefore, I believe this work should be considered for publication in Nature Communications; however, I believe there should be several points to address before the manuscript is published.*

Response: We thank the reviewer for commenting on the importance of this problem, and for being cautiously positive on the manuscript.

2. *As the authors note, partial differential equation constrained optimization has been explored by other authors before but only in model/test systems in 1-dimension. In many ways, the difficulty in the inverse problem is the numerical solution. The Wu and Yang approach with regularization should provide an ‘optimal’ solution and in relatively simple systems one can numerically find that solution. However, when one tries to apply it to large systems, then it becomes numerical intractable. Therefore, my largest concern about this work, is that this approach works great for small systems, but is still numerically intractable to any sizeable system. The largest systems presented here are 10 electrons. I think to showcase that this is a significant step forward, they authors should provide numerical results for a large system (50-100 electrons). If one can still solve the inverse problem using this partial differential equation constrained optimization methodology, it will really show that this work is a breakthrough in this field.*

Response: We thank the referee for suggesting that we demonstrate our approach on larger systems. In the revised manuscript, we have demonstrated the efficacy of our approach using two additional systems—(a) LDA density based verification for 1,3-dimethylbenzene (C_8H_{10} , 58 electrons), (b) CI density based exact v_{xc} evaluation for *ortho*-benzyne radical (C_6H_4 , 40 electrons), a strongly correlated system. In both the cases, we obtain similar accuracies as the smaller systems reported in this work. Accurately solving the inverse DFT problem on such large systems is unprecedented. We hope these additional results demonstrates the robustness and efficacy of the proposed approach.

3. *As stated above, it is the numerical difficulty which really makes this problem hard. Therefore, how is the convergence of these equations? Specifically, on convergence of $\|\rho_{data} - \rho\|_{L^2}$: (a) How many cycles does the density take to converge? (b) How is the convergence affected by the initial V_{xc} guess? (c) How is the convergence affected by the number of electrons in the system? (d) How is convergence affected by the weak, eg $H2(eq)$, and strong correlation, eg $H2(d)$, systems?*

Response:

- (a) For all the systems, except $\text{H}_2(d)$, it took 300-500 BFGS iterations to obtain an accuracy of $\|\rho_{data} - \rho\|_{L^2} < 10^{-4}$.
 - (b) We have experimented with two different initial guess—PW92 (LDA) potential and LB94 (GGA) potential. While the final result is independent of the initial guess, the LB94 guess takes $\sim 50 - 100$ lesser BFGS iterations to converge, as compared to the PW92 guess.
 - (c) We observe ~ 2 -fold increase in the number of iterations for H_2O , as compared to $\text{H}_2(eq)$ and $\text{H}_2(2eq)$, thereby suggesting that the rate of convergence is only weakly proportional to the the number of electrons.
 - (d) The $\text{H}_2(2eq)$ molecule, despite being a strongly correlated system, takes roughly the same number of iterations as that of $\text{H}_2(eq)$. However, for $\text{H}_2(d)$, we required $7 - 8\times$ the number of iterations as that of $\text{H}_2(eq)$. This slow convergence can be attributed to the fact that the Kohn-Sham HOMO-LUMO gap for $\text{H}_2(d)$ is ~ 3 mHa. Thus, from our observation, more than the nature of electronic correlations, the HOMO-LUMO gap governs the rate of convergence.
4. *In the H_2 stretched/dissociated case, what is the orbital energy of the LUMO going to? Are the significant difficulties in solving the strong correlation system with multiple near-degenerate orbitals? Naively, I would have expected that during the optimization cycles you could have potentials, which make the original LUMO become the HOMO as you try to force the strong-correlation solution. Does the optimization process continue to be well behaved?*

Response: For $\text{H}_2(2eq)$, the LUMO level is -0.311 Ha, with the HOMO level being -0.482 Ha. For the $\text{H}_2(d)$, the LUMO level is -0.476 Ha, with the HOMO level being -0.479 Ha. As correctly noted by the reviewer, this near-degeneracy in $\text{H}_2(d)$ affects the rate of convergence—it takes $7 - 8\times$ more iterations in comparison to $\text{H}_2(eq)$. However, we didn't notice any swapping of HOMO and LUMO between iterations, i.e., the HOMO always exhibited a bonding character (peaks on the two atoms having

similar signs) and the LUMO exhibited an anti-bonding character (oppositely signed peaks on the two atoms). Overall, although the optimization process had a slow rate of convergence, it remained stable and well behaved.

5. *Again, I want to emphasize that I believe this is important work. However, there have been several inverse methods that have been proposed over the last 15 years that have been applied to small systems that work relatively well. The difficulty in the problem is in the numerics and I think the authors should discuss and show that this optimization scheme is numerically tractable and simple to be able to deal with large and complex systems.*

Response: We thank the reviewer for being cautiously positive on this work. We hope the additional results on C₈H₁₀ and C₆H₄ (strongly correlated radical) demonstrates the efficacy of the proposed approach to deal with large complex systems.

Additional Changes

As part of this revision, we conducted inverse DFT simulations with better FE discretization for the atomic systems in the verification study (results in Figure 2 of the manuscript), which further improved the agreement between the v_{xc} obtained from inverse DFT calculation and $v_{xc}^{LDA}[\rho_{data}]$. The excellent agreement between the inverted v_{xc} and $v_{xc}^{LDA}[\rho_{data}]$ is also evident from Figure 2 in SI, which shows the data on a log scale to demonstrate the near-field and far-field agreement.

References

- [1] Felix Brockherde, Leslie Vogt, Li Li, Mark E Tuckerman, Kieron Burke, and Klaus-Robert Müller. Bypassing the Kohn-Sham equations with machine learning. *Nature Commun.*, 8(1):872, 2017.

- [2] Ryo Nagai, Ryosuke Akashi, Shu Sasaki, and Shinji Tsuneyuki. Neural-network Kohn-Sham exchange-correlation potential and its out-of-training transferability. *The Journal of chemical physics*, 148(24):241737, 2018.
- [3] Robert van Leeuwen and Evert Jan Baerends. Energy expressions in density-functional theory using line integrals. *Physical Review A*, 51(1):170, 1995.
- [4] Christoph R Jacob. Unambiguous optimization of effective potentials in finite basis sets. *J. Chem. Phys.*, 135(24):244102, 2011.
- [5] Tim Heaton-Burgess, Felipe A. Bulat, and Weitao Yang. Optimized effective potentials in finite basis sets. *Phys. Rev. Lett.*, 98:256401, 2007.
- [6] Felipe A. Bulat, Tim Heaton-Burgess, Aron J. Cohen, and Weitao Yang. Optimized effective potentials from electron densities in finite basis sets. *J. Chem. Phys.*, 127(17):174101, 2007.
- [7] Qin Wu and Weitao Yang. A direct optimization method for calculating density functionals and exchange-correlation potentials from electron densities. *J. Chem. Phys.*, 118(6):2498–2509, 2003.
- [8] P. R. T. Schipper, O. V. Gritsenko, and E. J. Baerends. Kohn-Sham potentials corresponding to slater and gaussian basis set densities. *Theor. Chem. Acc.*, 98(1):16–24, 1997.
- [9] Alex P. Gaiduk, Ilya G. Ryabinkin, and Viktor N. Staroverov. Removal of basis-set artifacts in Kohn-Sham potentials recovered from electron densities. *J. Chem. Theory Comput.*, 9(9):3959–3964, 2013.

Reviewers' comments:

Reviewer #1 (Remarks to the Author):

In my view, all referee comments have been adequately addressed. For completeness, I'd like to remark that inverse DFT problems have been solved for systems of comparable (or slightly larger) size as the new examples here in different contexts like reconstructing embedding potentials (e.g., JCP 149, 054103). But the accuracy obtained in those applications is significantly lower, and of course the reference densities are of lower quality as well. In that sense, I'd agree to the statement that the new examples are unprecedented system sizes for inverse DFT calculations of such high accuracy.

Reviewer #2 (Remarks to the Author):

The minor revisions and clarifications made by the authors do not change my assessment: the proposed method is neither more robust nor more accurate than many existing approaches to DFT inversion. For non-Gaussian densities, any careful numerical inversion procedure would give as good potentials as those of Fig. 2. For Gaussian densities, the potentials of this work are still imperfect (crude and wobbly) - see Figs. 6 and 7. Moreover, what makes the method work passably for Gaussian densities is not the proposed constrained minimization of Eq. 5 but an expedient alteration of the input density.

The authors are billing their method as "an advance that overcomes all the aforementioned open issues in the inverse DFT problem", "a novel finite-element basis approach that is systematically convergent and complete, and thereby eliminates the ill-conditioning associated with previous approaches". However, they do not provide a single direct comparison with any of the previous approaches to substantiate those claims. In the absence of such evidence, the claims are not ready for publication.

Reviewer #3 (Remarks to the Author):

In the revised manuscript the authors have addressed the concerns I brought up in the original manuscript. The application to significantly larger systems shows the robustness of the method and the ability to perform these inverse calculations on complicated systems that will be of interest to the community.

Given the number of years people have worked on this problem and these authors have provided a relatively simple to implement and accurate method, I recommend publishing this manuscript.

My only comment to the authors about improving the manuscript would be to include the information about convergence that was provided in the response to the reviewer. I found this information very helpful. Particularly the discussion about how the HOMO-LUMO gap affects the convergence. I think other readers would enjoy this information in the main text of SI.

Response to Reviewers

Bikash Kanungo, Paul M. Zimmerman, Vikram Gavini

We thank the reviewers for the careful review of the manuscript, and their comments and suggestions to improve the manuscript. Below we are providing a detailed response to the reviewers' comments and the changes made to the manuscript. We hope we have satisfactorily addressed all the comments of the reviewers.

Response to Reviewer #1

In my view, all referee comments have been adequately addressed. For completeness, I'd like to remark that inverse DFT problems have been solved for systems of comparable (or slightly larger) size as the new examples here in different contexts like reconstructing embedding potentials (e.g., JCP 149, 054103). But the accuracy obtained in those applications is significantly lower, and of course the reference densities are of lower quality as well. In that sense, I'd agree to the statement that the new examples are unprecedented system sizes for inverse DFT calculations of such high accuracy.

Response: We thank the Reviewer for the careful review of our manuscript. We are glad that, in the Reviewer's opinion, we have addressed all the referee comments adequately.

Response to Reviewer #2

The minor revisions and clarifications made by the authors do not change my assessment: the proposed method is neither more robust nor more accurate than many existing approaches to DFT inversion.

Response: We thank the reviewer for the comments, but respectfully disagree on the assessment of our work. We note that comments from Reviewer 1 and Reviewer 3 underscore the robustness of the proposed method in accurately solving the inverse DFT problem on large complicated systems that will be of interest to the community. In the SI of the revised manuscript, we have added a section that compares the proposed approach to existing approaches in terms of accuracy, robustness and computational viability. Below we provide our response to the specific comments made by the reviewer.

1. *For non-Gaussian densities, any careful numerical inversion procedure would give as good potentials as those of Fig. 2.*

Response: We agree with the the reviewer that for non-Gaussian densities (eg., Slater basis-set densities) one can obtain accurate v_{xc} 's through a careful numerical inversion. However, this is contingent upon the use of a complete basis for discretizing both the Kohn-Sham orbitals ψ_i 's and the v_{xc} , without which the inversion suffers from non-unique or oscillatory potentials [1, 2, 3]. This vitality of the completeness of the basis, even while using non-Gaussian basis-set densities, have been noted in [4, 5, 6, 7]—the only available literature to use non-Gaussian basis-set densities (Slater densities or densities on a radial grid) along with a numerical grid (to achieve completeness). However, we emphasize that all the calculations presented in these works are for 1D (atomic) systems, and an extension of their numerical strategy to 3D systems remains computationally prohibitive. To this end, the use of finite-elements, as employed in this work, provides both the required completeness and computational efficiency to handle 3D systems.

2. *For Gaussian densities, the potentials of this work are still imperfect (crude and wobbly)*

- see Figs. 6 and 7

Response: We agree with the reviewer that it is desirable to obtain a smoother v_{xc} for the dissociated hydrogen molecule ($H_2(d)$) than reported in this work. However, we re-emphasize that $H_2(d)$ is a notably challenging system for inverse DFT, and this work remains the first attempt to accurately evaluate the exact v_{xc} for a dissociated molecule. The slight wobbliness is a consequence of the incorrect asymptotics of the Gaussian basis-set density. More importantly, we highlight that the v_{xc} 's obtained for all other systems are remarkably smooth.

3. *Moreover, what makes the method work passably for Gaussian densities is not the proposed constrained minimization of Eq. 5 but an expedient alteration of the input density.*

Response: We agree with the reviewer that the use of $\Delta\rho$ correction to the CI density is an important aspect our method. This in itself, however, is not sufficient to attain accurate v_{xc} 's. One still needs to use a complete basis to obtain accurate and smooth v_{xc} 's. We illustrate this in Fig. 1, which provides the v_{xc} for $H_2(eq)$ corresponding to a coarser finite-element mesh than that was used for the results presented in the main manuscript. As evident, the lack of sufficient resolution of the basis in the coarse mesh results in oscillations around the atoms. This underlines the fact that even with the correct density, the use of a complete basis is indispensable to obtaining accurate v_{xc} 's.

4. *However, they do not provide a single direct comparison with any of the previous approaches to substantiate those claims. In the absence of such evidence, the claims are not ready for publication.*

Response: We agree with the reviewer that a direct comparison with previous approaches can provide a more objective assessment of our findings. However, given the importance of the inverse DFT problem, many attempts have been made at solving this problem over the past 20 years. A direct comparison with all the proposed approaches is challenging, as re-implementing these methods (due to non-availability of codes) and resolving the issues there-within, especially for 3D molecular systems, is a

time-consuming task. More importantly, the performance of these approaches and the outstanding issues are well documented in the literature.

Thus, in the revised manuscript, we have used the data and results from the literature to compare and contrast existing methods for DFT inversion with our proposed approach, in terms of accuracy, robustness and computational viability. A new section in the SI (Section 5) is devoted to this comparison. Further, we have made appropriate revisions to ensure that all the claims in the revised manuscript are supported by our objective findings.

Response to Reviewer #3

In the revised manuscript the authors have addressed the concerns I brought up in the original manuscript. The application to significantly larger systems shows the robustness of the method and the ability to perform these inverse calculations on complicated systems that will be of interest to the community. Given the number of years people have worked on this problem and these authors have provided a relatively simple to implement and accurate method, I recommend publishing this manuscript.

Response: We thank the reviewer for the careful review of the manuscript. We are glad that we have addressed all the reviewer's concerns satisfactorily.

My only comment to the authors about improving the manuscript would be to include the information about convergence that was provided in the response to the reviewer. I found this information very helpful. Particularly the discussion about how the HOMO-LUMO gap affects the convergence. I think other readers would enjoy this information in the main text of SI.

Response: In the revised manuscript, we have added a section in the SI discussing the convergence aspects, and its dependence on the HOMO-LUMO gap.

References

- [1] Christoph R Jacob. Unambiguous optimization of effective potentials in finite basis sets. *J. Chem. Phys.*, 135(24):244102, 2011.
- [2] Tim Heaton-Burgess, Felipe A. Bulat, and Weitao Yang. Optimized effective potentials in finite basis sets. *Phys. Rev. Lett.*, 98:256401, 2007.
- [3] Felipe A. Bulat, Tim Heaton-Burgess, Aron J. Cohen, and Weitao Yang. Optimized effective potentials from electron densities in finite basis sets. *J. Chem. Phys.*, 127(17):174101, 2007.
- [4] R. van Leeuwen and E. J. Baerends. Exchange-correlation potential with correct asymptotic behavior. *Phys. Rev. A*, 49:2421–2431, 1994.
- [5] P. R. T. Schipper, O. V. Gritsenko, and E. J. Baerends. Kohn-Sham potentials corresponding to slater and gaussian basis set densities. *Theor. Chem. Acc.*, 98(1):16–24, 1997.
- [6] Karel Peirs, Dimitri Van Neck, and Michel Waroquier. Algorithm to derive exact exchange-correlation potentials from correlated densities in atoms. *Phys. Rev. A*, 67(1):012505, 2003.
- [7] Ilya G. Ryabinkin and Viktor N. Staroverov. Determination of KohnSham effective potentials from electron densities using the differential virial theorem. *J. Chem. Phys.*, 137(16):164113, 2012.

(a)

(b)

Figure 1: The $v_{xc}(\mathbf{r})$ for $H_2(eq)$ for: (a) fine mesh (used for results in the main manuscript), and (b) a coarse mesh.